# HANDLE: Robust Non-prehensile Liquid Manipulation in Cluttered, Human-centric Environments with Physical HRI

Anonymous

*Abstract*—Robust manipulation in human-centered environments requires handling tightly coupled sources of uncertainty, including dynamic obstacles, physical human interaction, and payload-induced constraints. This challenge is particularly acute in non-prehensile liquid transport, where external disturbances directly affect both task feasibility and system stability. We present **HANDLE (Human-Aware Non-prehensile hanDling of Liquids with whole-body rEactive Control)**, a real-time framework for robust manipulation under coupled dynamic and interaction disturbances. **HANDLE employs a hierarchical robustness architecture that explicitly addresses three complementary aspects of robustness: (i) environmental robustness via a geometric-aware safety layer enabling real-time whole-body collision avoidance in dynamic and cluttered scenes; (ii) dynamic robustness via a slosh-aware control layer that regulates accelerations and orientation to maintain fluid stability; and (iii) interaction robustness via a compliance mechanism that accommodates direct human-applied forces without compromising previous conditions. Central to our approach is a constraint-consistent projection mechanism that maps arbitrary external inputs—including teleoperation commands and stochastic human disturbances—onto a stability-constrained safe manifold, ensuring feasibility without sacrificing responsiveness. We evaluate the proposed framework under aggressive and unpredictable conditions, including abrupt velocity commands, strong physical human guidance, and simultaneous multi-modal physical contacts and disturbances in shared workspaces. Results demonstrate that HANDLE maintains constraint satisfaction, prevents spillage and collisions, and preserves stable non-prehensile transport under conditions where state-of-the-art methods fail. A user study further indicates improved perceived safety, robustness, and operational intuitiveness [1].**

*Index Terms*—Robust Robot Manipulation, Human–Robot Interaction, Physical Human Guidance, Disturbance Rejection, Constraint-Based Control, Whole-Body Reactive Control, Slosh-Free Transport

## I. INTRODUCTION

Robotics have seen rapid progress in recent years, transitioning from structured workcells into human-centered and unstructured environments. However, achieving robust manipulation in such settings remains a fundamental challenge, as robots must operate under tightly coupled sources of uncertainty, including dynamic obstacles, human interaction, and task-dependent physical constraints. A representative example is non-prehensile liquid transportation in cluttered and dynamic environments, particularly in close proximity to humans. This task is inherently sensitive to disturbances, as even small external perturbations can induce sloshing, leading to spillage or task failure. Indeed, studies on phenomena such as coffee spilling have shown that successful transport depends on a complex coupling between hand motion and fluid dynamics, influenced by container geometry, excitation frequency, and

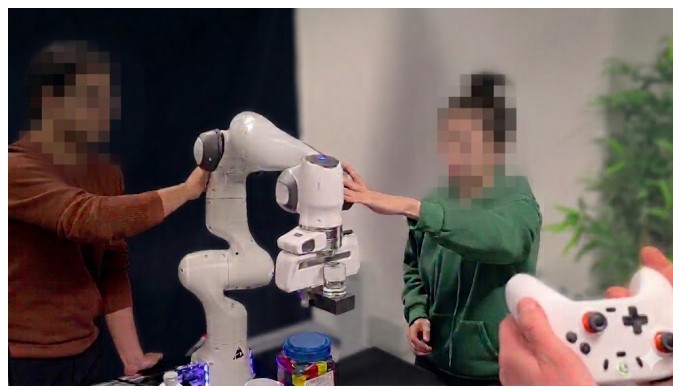

Fig. 1: The HANDLE framework in heterogeneous HRI scenarios enforces multi-layer robustness for non-prehensile liquid transportation by unifying robot–liquid dynamics, coexistence robustness, and physical compliance. *Robot–Liquid Dynamics Robustness:* The framework regulates robot motion under joint and manipulability constraints while adaptively adjusting the end-effector to maintain slosh-free and stable transport under disturbances. *Coexistence Robustness:* It ensures safe clearance between humans, the robot, and obstacles via an analytically defined reactive geometric model for dense environments. *Physical Compliance:* The controller enables compliant multi-point physical interaction, integrating external forces while preserving coexistence safety and liquid stability.

whole-body coordination [1], [2]. To mitigate these effects, humans often rely on external aids such as lids, specialized containers, or trays[2]—effectively offloading robustness to mechanical design rather than control [2], [5]. Robots, in contrast, struggle with manipulating complex external tools, while they excel in high accuracy and repeatability, making them well-suited for regulating desired motion behaviours even without external aids. This has also been demonstrated for slosh-free liquid transportatio [5], [6].However, these approaches largely assume controlled environments and limited disturbances, and thus fail to generalize to real-world settings characterized by dynamic obstacles, human interaction, and unpredictable external inputs. As a result, enabling robust liquid manipulation under such conditions remains an open challenge. From an HRI perspective, we interpret robustness as a multi-layered robustness requirement under coupled disturbances: (i) *Intrinsic robustness*, ensuring dynamically consistent robot behavior to maintain stability under disturbance-sensitive liquid dynamics and intrinsic robot constraints; (ii) *Coexistence robustness*, ensuring robust collision avoidance with humans and obstacles through continuous reactive adaptation; (iii) *Physical Compliance*, enabling stable and predictable responses

to direct human-applied forces. Importantly, these requirements are enforced at the whole-body level rather than being limited to the end-effector, such that all joints and links contribute to maintaining constraint consistency during interaction. However, these objectives are inherently coupled and often conflicting, particularly under external disturbances. For instance, increasing responsiveness to dynamic obstacles may reduce compliance to human-applied forces, while enforcing strict dynamic consistency to prevent sloshing can limit the system's ability to react to sudden environmental or interaction changes. As a result, robustness in such settings requires maintaining feasibility under simultaneously active and competing constraints in real time.

[2]The most successful being the Moroccan tea tray device [3], and its modern version, the spillnot [4], whose dynamics have inspired the current work.

To address this challenge, we propose a hierarchical framework, **HANDLE** — *H*uman-*A*ware *N*on-prehensile han*D*ling of *L*iquids with whole-body r*E*active Control — which unifies these objectives within a single real-time control architecture. HANDLE ensures robustness by projecting external inputs—including teleoperation commands and human-applied forces—onto a stability-constrained feasible manifold, guaranteeing constraint satisfaction without sacrificing responsiveness. The proposed framework enables robots to perform disturbance-sensitive manipulation tasks in a safe, predictable, and robust manner by jointly accounting for interaction constraints and liquid dynamics. We evaluate the approach under realistic and adversarial conditions, including teleoperation, direct physical human interaction, and operation in cluttered, dynamically changing environments. Crucially, these factors are not treated in isolation but are handled simultaneously within a unified framework, demonstrating robust performance under multi-modal disturbances. Fig. 1 illustrates such scenarios.

**Contributions**

HANDLE is a unified hierarchical framework for real-time, disturbance-resilient manipulation, enabling slosh-free liquid transport under coupled dynamic, environmental, and interaction disturbances. The framework jointly considers whole-body robot motion and enforces: **(i)** dynamic robustness through slosh-free consistency and intrinsic stability, **(ii)** environmental robustness via collision-free operation in cluttered human-centered environments, and **(iii)** interaction robustness through compliant behavior under multi-contact physical human–robot interaction. To achieve this, HANDLE models the robot, human, and environment using analytical geometric primitives, enabling efficient geometric-aware robustness filtering via control barrier functions. External inputs—including teleoperation commands and human-applied forces—are projected onto a stability-constrained feasible manifold, ensuring constraint consistency without sacrificing responsiveness. At execution time, the framework generates feasible whole-body motions that remain robust to disturbances while preserving task-level constraints. Together, these mechanisms enable operation near performance limits while maintaining robustness and task feasibility.

The main contributions are summarized as follows:

*a) Geometric-aware robustness for human–environment interaction:* A real-time robustness filtering layer based on analytical geometric representations that ensures whole-body collision avoidance under dynamic and uncertain environments, enabling non-conservative and robust motion adaptation.

*b) Disturbance-rejected physical human–robot interaction:* The first framework to explicitly integrate direct pHRI into

non-prehensile liquid transport while preserving task feasibility, by projecting human-applied forces onto slosh-free constraint sets and enabling compliant whole-body adaptation under external perturbations.

*c) Unified constraint control under coupled disturbances:* A hierarchical control architecture that jointly enforces environmental robustness, interaction compliance, and liquid dynamics constraints, maintaining feasibility under conflicting objectives and unpredictable inputs in real time.

*d) Robustness validation under multi-modal disturbances::* Extensive experiments in cluttered, dynamic, and human-interactive environments, including aggressive teleoperation and physical guidance, demonstrating consistent task execution, disturbance rejection, and improved user-perceived robustness.

These contributions advance the state of the art in robust robot manipulation by demonstrating that constraint-consistent control enables reliable execution of disturbance-sensitive tasks in complex and unpredictable human environments.

## II. PROBLEM FORMULATION & HANDLE ARCHITECTURE

Our goal is to realize robust non-prehensile liquid manipulation in heterogeneous pHRI, allowing responsive behaviour to dynamic obstacles, teleoperation commands, and direct physical human guidance. At each control cycle, the controller receives a Cartesian reference velocity $\boldsymbol{v}_{\text{ref}} \in \mathbb{R}^3$, generated either by an autonomous planner or by a human operator, together with a runtime set of geometric primitives $\mathcal{G}(t)$ describing nearby obstacles and human skeleton parts. The controller seeks a real-time mapping

$$\pi : (\boldsymbol{q}, \boldsymbol{v}_{\text{ref}}, \mathcal{G}(t), \boldsymbol{\tau}_{\text{ext}}, \boldsymbol{F}_{\text{ext}}) \mapsto (\boldsymbol{\xi}^\star, \dot{\boldsymbol{q}}^\star), \qquad (1)$$

where $\boldsymbol{q}$ is the robot configuration, $\boldsymbol{F}_{\text{ext}}$ and $\boldsymbol{\tau}_{\text{ext}}$ denote external task-space and joint-space interaction inputs, and $\boldsymbol{\xi}^\star = [\boldsymbol{v}^{\star\top} \ \boldsymbol{\omega}^{\star\top}]^\top$ is the executed task-space twist.

HANDLE interprets robustness as the simultaneous satisfaction of three coupled conditions. *Environmental robustness* requires whole-body collision-free coexistence with humans and obstacles. This is captured by the geometric admissible set

$$\mathcal{C}_{\text{geo}}(t) = \bigcap_i \{ \boldsymbol{q} \mid h_i(\boldsymbol{q}, t) \geq 0 \}, \qquad (2)$$

where each barrier function $h_i$ encodes a signed-distance margin between robot and external geometric primitives [7]. Second, *robot-intrinsic feasibility* requires the motion to remain feasible with respect to joint limits and manipulability, yielding the set $\mathcal{C}_{\text{robot}}$. Third, *slosh-free orientation conditions* requires the inertial frame depicting the transported liquid to remain close to the slosh-free condition. Following the pendulum-based slosh-free formulation in [5], [6], the desired tilt satisfies $(a_z + g)\tan\theta = a_{xy}$, where $a_{xy}$ and $a_z$ are the lateral and vertical accelerations of the pivot. The corresponding slosh-free set is denoted by $\mathcal{C}_{\text{SFS}}$. Together, these sets define the robustness envelope within which HANDLE must operate.

Rather than solving a single monolithic optimization, HANDLE exploits the hierarchical structure shown in Fig. 2. The *outer loop* enforces environmental awareness and robustness by projecting the commanded translational motion onto $\mathcal{C}_{\text{geo}}$ through a CBF-based quadratic program,

$$\boldsymbol{v}_{\text{geo}} = \arg\min_{\boldsymbol{v}} \frac{1}{2}\|\boldsymbol{v} - \boldsymbol{v}_{\text{ref}}\|^2 \text{ s.t. } \nabla h_i(\boldsymbol{q}, t)^\top \boldsymbol{v} + \alpha h_i(\boldsymbol{q}, t) \geq 0, \forall i,$$

thereby generating the closest coexistence-feasible motion to the command. In parallel, end-effector (EE) physical interaction

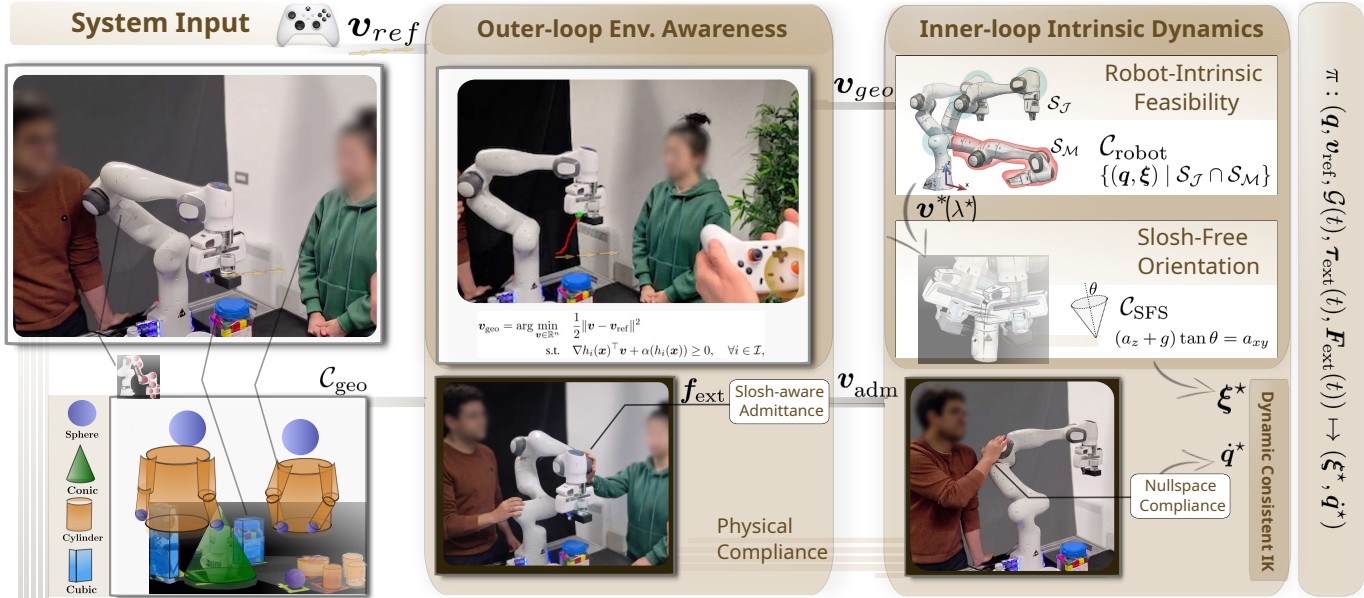

Fig. 2: **The HANDLE Pipeline**. The proposed framework consists of an outer-loop environment awareness layer and an inner-loop intrinsic dynamics layer. The reference velocity $\boldsymbol{v}_{ref}$ is first processed by a geometric co-existence robustness module, which ensures safe robot–environment interaction via a CBF-based QP formulation, producing $\boldsymbol{v}_{geo}$. The motion is then refined by the robot intrinsic dynamics layer, which enforces joint and motion constraints to compute a dynamically feasible velocity $\boldsymbol{v}^*$. Next, a liquid robustness module introduces an orientation controller with finite-horizon prediction, optimizing end-effector tilting and generating the augmented command $\boldsymbol{\xi}^* = [\boldsymbol{v}^*, \boldsymbol{\omega}^*]$ for slosh-free liquid transportation. Meanwhile, physical compliance is decomposed into Cartesian space and robot nullspace. In task space, an admittance-based pHRI controller produces $\boldsymbol{v}_{adm}$, which is combined with $\boldsymbol{v}{geo}$ as input to the inner loop. Finally, task-space commands are mapped to joint space via a dynamically consistent IK solver and a joint nullspace admittance controller. reduce the caption

is incorporated through an admittance term $\boldsymbol{v}_{\mathrm{adm}}$, producing the interaction-aware command $\boldsymbol{v}_{\mathrm{int}} = \boldsymbol{v}_{\mathrm{geo}} + \boldsymbol{v}_{\mathrm{adm}}$. Thus, the outer loop resolves obstacle avoidance and task-space human guidance at the translational level before intrinsic dynamics and slosh-free-specific constraints are enforced within the orientation and time domains.

The *inner loop* refines $\boldsymbol{v}_{\mathrm{int}}$ through two nested stages. First, a scalar time-to-bound filter enforces robot-intrinsic feasibility by scaling the translational command according to the remaining margins to joint and manipulability boundaries, as in the slosh-free shared-autonomy framework of [6]. Concretely,

$$\lambda^\star = \arg \max_{\lambda \in [0,1]} \lambda \text{ s.t. } \Delta t_{\mathrm{lim}} \leq \Delta t_B^J/\lambda, \quad \Delta t_{\mathrm{lim}} \leq \Delta t_B^M/\lambda,$$

where $\Delta t_B^J/\lambda$ and $\Delta t_B^M/\lambda$ denote the predicted times to the nearest joint-limit and manipulability boundaries under the motion stemming from the outer-loop. The resulting feasible command is $\boldsymbol{v}^\star = \lambda^\star \boldsymbol{v}_{\mathrm{int}}$. Second, $\boldsymbol{v}^\star$ is treated as the nominal pivot motion of a reduced spherical-pendulum model. Using the pivot acceleration input, $\boldsymbol{u} = \begin{bmatrix} \ddot{x}_p & \ddot{y}_p & \ddot{z}_p \end{bmatrix}^\top$, the linearized tilt dynamics are

$$l\ddot{\theta} = -g\theta + u_1, \qquad l\ddot{\phi} = -g\phi - u_2,$$

which are discretized and embedded in a finite-horizon MPC. The MPC computes the first admissible control input and the associated tilt-consistent orientation profile, yielding the full task-space command $\boldsymbol{\xi}^\star = [\boldsymbol{v}^{\star\top} \ \boldsymbol{\omega}^{\star\top}]^\top$.

Finally, the execution layer maps the command $\boldsymbol{\xi}^\star$ to joint space through dynamically consistent inverse kinematics and adds null-space compliance so that redundant links remain responsive to direct physical interaction without disturbing the primary transport task,

$$\dot{\boldsymbol{q}}^\star = \boldsymbol{J}_M^\dagger \boldsymbol{\xi}^\star + (\boldsymbol{I} - \boldsymbol{J}_M^\dagger \boldsymbol{J})\dot{\boldsymbol{q}}_{\mathrm{adm}}, \tag{3}$$

where $\dot{\boldsymbol{q}}_{\mathrm{adm}}$ is a saturated joint-space whole-body admittance controller.

In this way, HANDLE distributes robustness across layers rather than treating it as a single property: the outer loop handles environmental and interaction-aware motion generation, while the inner loop preserves robot feasibility and slosh-free transport. This decomposition is what enables the system to remain responsive under coupled disturbances while preserving task-level consistency.

## III. EXPERIMENTAL VALIDATION

We evaluate HANDLE on a Franka Emika Panda manipulator in simulation and real-world settings, with emphasis on robustness under coupled environmental, dynamic, and interaction disturbances. Our evaluation follows three levels. **(i) Environmental robustness:** a component-wise analysis of the geometric-aware robustness layer is provided separately in **Appendix IV-B**, where we study its computational efficiency, motion quality, and robustness to dynamic obstacle configurations. **(ii) Multi-modal disturbance rejection:** Beyond obstacle avoidance for environmental robustness in (i), we evaluate the fully integrated system under teleoperation (Fig. 3) and physical contact (Fig. 4), considering their combined multi-modal effects (Fig. 4). **(iii) Human-centred robustness assessment:** we conduct a within-subject user study to evaluate perceived ease of operation, safety, physical effort, and overall acceptance of the full framework.

*a) Slosh-Free Dynamic Transport under Disturbances:* First, we consider an experiment with a joystick teleoperation in cluttered scenes, where the commanded motion contains abrupt changes in direction and speed. As shown in Fig.3, HANDLE reshapes the raw translational command through the hierarchical controller and maintains a tilt-consistent orientation of the container. This preserves the slosh-free transport

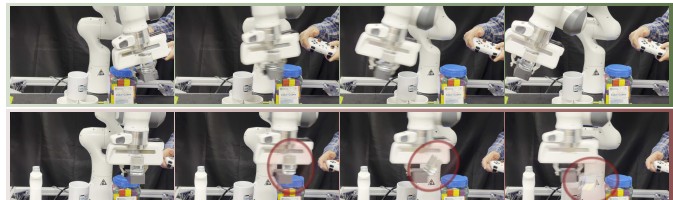

Fig. 3: **Teleoperated Non-prehensile Manipulation. Top**: Disturbance rejected motion. **Bottom**: Motion without disturbance rejection.

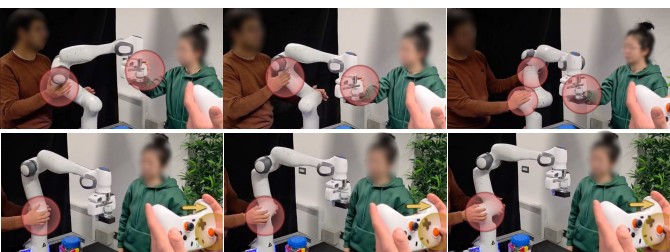

Fig. 4: **Non-prehensile Manipulation in Multi-Modal pHRI. Top:** Whole-body physical contact compliance. **Bottom: Multi-modal pHRI.** HANDLE integrates teleoperation, reactive avoidance, and physical contact, enabling robust disturbance rejection under simultaneous interaction. Yellow arrows indicate teleoperation commands.

condition even during aggressive maneuvers. In contrast, a baseline approach based on a min. jerk profile that keeps the EE orientation upright fails still induces lateral accelerations that destabilise the load , causing severe sloshing and eventual task failure. These experiments show that smooth motion alone is insufficient under disturbance-rich teleoperation; robustness requires explicit enforcement of the slosh-free condition.

We next evaluate the full system in a scenario combining teleoperation, human coexistence, and direct physical interaction. As illustrated in Fig. 4, the robot must simultaneously track a teleoperated command, avoid a nearby human, and remain compliant to body contact and manual guidance. HANDLE maintains this coupled behavior by continuously projecting the commanded motion onto the admissible set defined by the environmental, robot-intrinsic, and slosh-free constraints. The top sequence additionally shows that end-effector guidance and elbow perturbations can be separated into task-space and null-space responses, allowing whole-body compliance without disturbing the primary transport task. Together, these results validate the full pipeline under realistic, overlapping disturbance sources.

### A. Human-Centred Assessment

To evaluate the framework from the user perspective, we conducted a within-subject exploratory study with $n = 15$ participants, who experienced paired conditions corresponding to the presence or absence of the manipulation robustness, human/environmental robustness, and physical contact module. The order of conditions was randomized across participants. Subjective responses were collected on a 6-point Likert scale (0–5), and paired comparisons were analyzed with Wilcoxon signed-rank tests using Holm–Bonferroni correction. We report median [IQR] values together with effect sizes $r = |Z|/\sqrt{n}$.

The results in Fig. 5 show a consistent pattern across all three modules. Further details are given in Table III within Appendix V-B. For slosh-free conditions, participants rated the system substantially higher in perceived slosh-free safety,

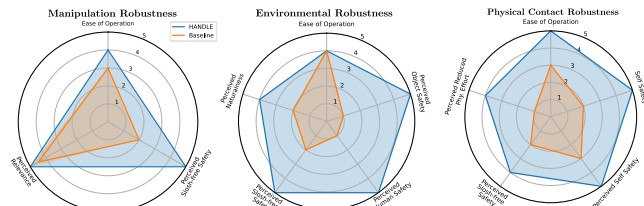

Fig. 5: **Multi-dimensional user-study of HANDLE.** The radar plots illustrate robustness across multiple aspects, including non-prehensive (slosh-free) task safety without compromising ease of operation. HANDLE improves perceived human and object safety during teleoperation, and enhances ease of operation, self-safety, and physical comfort during physical contact, demonstrating improved human preference and system robustness under multi-modal interactions.

while ease of operation remained unchanged. This indicates that the liquid-aware compensation improves perceived transport robustness without increasing control difficulty. The environment robustness layer strongly improved perceived human and object safety, again without reducing ease of operation. This suggests that whole-body reactive avoidance improves confidence during teleoperation without adding user burden. While for the physical contact layer, the framework increased ease of operation, improved perceived self-safety, and reduced perceived physical effort during direct guidance.

At the overall level, participants rated the full HANDLE framework higher in perceived motion naturalness than the baseline, and most participants reported positive adoption attitudes. Specifically, 14/15 agreed that the system was useful, and 10/15 agreed that it was necessary for HRI-based transportation tasks. These findings complement the qualitative experiments by showing that the same mechanisms that improve constraint-consistent robustness are also perceived positively by users during collaborative manipulation.

### IV. CONCLUSION AND FUTURE WORK

In this paper, we presented the HANDLE framework, a unified hierarchical approach designed to transition delicate manipulation tasks from controlled workcells into cluttered, human-populated environments. By integrating a geometric-aware safety layer with whole-body physical compliance, we addressed the fundamental trade-off between operational reactivity, human safety, and the dynamic robustness required for slosh-free transport.

Our results demonstrate that liquid transportation remains a highly sensitive task under external disturbances and human interaction, often too fragile for deployment outside controlled settings can be executed reliably under abrupt commands, direct physical guidance, and overlapping multi-modal disturbances. The synergy of the HANDLE modules ensure that the robot remains a safe, predictable, and compliant partner, capable of maintaining a stable, slosh-free state even under unexpected external perturbations or aggressive human guidance. The user study indicates that these robustness properties are not only enforced by the controller, but also perceived by users through improved naturalness, ease of operation, and confidence during interaction. Future work will focus on increasing robustness to sensing noise and environmental uncertainty, and on longitudinal human studies to evaluate how certified reactive transport influences trust, comfort, and psychological safety in real-world service scenarios such as hospitals and busy restaurants.

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

## V. Appendix

### A. Literature Review

Existing work on robotic liquid transportation spans a broad methodological spectrum, which can be categorized based on the fidelity of the underlying sloshing representation. At one end, high-fidelity fluid simulations and reduced-order models explicitly capture liquid dynamics through particle-based formulations, modal representations, or physically grounded approximations [8], [9]. These approaches enable accurate prediction and planning, but are typically computationally intensive and therefore limited to offline trajectory optimization or model-predictive frameworks [10], [11]. At the other end, surrogate low-order models provide tractable approximations that enable real-time control and shared autonomy, trading modeling accuracy for computational efficiency [6], [12]–[14].

Across both categories, methods can also be distinguished along the control axis, ranging from offline feedforward optimization [8]–[11], [15], [16] to closed-loop feedback strategies enabled by surrogate models [6], [12], [13]. A key insight across this literature is that slosh suppression and slosh-free conditions are governed by regulating the effective gravito-inertial forces acting on the liquid. However, most existing approaches implicitly assume nominal conditions and do not explicitly address robustness to external disturbances, such as human interaction, environmental changes, or unmodeled dynamics.

While smooth trajectories and acceleration regulation can mitigate sloshing, they are generally insufficient under disturbance-rich conditions. Moreover, simply reducing accelerations often degrades system performance. Effective slosh-free motion requires actively compensating for time-varying gravito-inertial effects through orientation adaptation, either via explicit modeling or orientation-aware planning [17]. Recent work has therefore explored constraint-based formulations and data-driven models to balance real-time feasibility and slosh-free motion [5], [18], [19]. Shared teleoperation strategies have further extended these ideas to interactive settings [20]. Nevertheless, these methods remain limited in their ability to handle fast, unpredictable disturbances, due to finite controller response speeds and simplified contact models (e.g., coarse friction cone approximations). Crucially, they do not provide guarantees under direct human-applied forces or multi-modal interaction.

More broadly, most liquid manipulation approaches remain confined to controlled or weakly varying environments. Whether in autonomous execution [5], [16], [18] or teleoperation [6], [21], [22], existing methods primarily regulate end-effector motion in structured or free-space settings. As a result, they lack robustness to dynamic obstacles, physical interaction, and whole-body constraints required for operation in realistic human-centered environments.

In parallel, reactive motion generation in cluttered and dynamic environments has been extensively studied in robotics. Established approaches include artificial potential fields [13], [23]–[25], vector-field inequalities [26], optimization-based robustness filters [27]–[29], and control barrier functions (CBFs) [30], [31]. These methods provide strong guarantees for collision avoidance and have been extended to certain forms of safe physical human–robot interaction [32]–[35].

However, these approaches largely focus on free-motion tasks and do not account for disturbance-sensitive payload dynamics. In liquid and non-prehensile manipulation, reactive motions that are safe from a collision-avoidance perspective may still violate task feasibility due to induced lateral forces. This reveals a fundamental robustness gap: robustness with respect to the environment does not imply robustness with respect to the task dynamics. For example, abrupt deceleration near obstacles, joint-limit avoidance maneuvers, or reactive reorientation can introduce conflicting constraints between collision avoidance, physical feasibility, and liquid stability. This challenge is further exacerbated in physical human–robot interaction, where the system must remain simultaneously reactive, compliant, and dynamically consistent under external disturbances.

### B. Experiment Validation: Environment robustness Component-wise Quantitative Analysis

We first validate the outer-loop downstream effects on motion quality. In these trials, we deploy a reactive trajectory generator based on an attractor formulation with constraint handling inspired by [24], [25]. It is important to emphasize that this module serves solely as a reference motion generator for the HANDLE framework and is not tied to any specific implementation. Alternative reactive planning paradigms can also be employed, e.g., potential field methods [36], trajectory interpolation approaches [37], and diffusion-based motion generation [38].

In this framework, the target position generates an attractive force that guides the manipulator, while robustness is enforced through control barrier function (CBF) constraints. This setup allows us to evaluate how an attraction-driven trajectory is reshaped under different scene representations to ensure collision avoidance and be robust under complex scenarios. We compare three scene representation strategies: (i) GEO, our proposed approach based on analytical geometric primitives with smooth distance gradients; (ii) BBOX, a baseline that represents obstacles using axis-aligned bounding boxes; and (iii) P.CLOUDS, which models obstacles as a discrete set of points/spheres, discretized at a resolution of $0.01$ m. The impact of these representations on the resulting trajectories is illustrated in Fig. 6.

*a) Computational Efficiency and Obstacle Clearance:* The results summarized in Table I demonstrate that our GEO method significantly outperforms analytical BBOX and P. CLOUDS. across all geometric primitives. Regarding *calculation time*, GEO achieves a per-object computation time of approximately $0.0018$–$0.0027$ ms for complex primitives like cones and cylinders, which is nearly 5 times faster than the BBOX baseline ($\approx 0.01$ ms) and 10 times faster than P.CLOUDS ($\approx 0.022$ ms). Even for cubes, where the geometry is identical, GEO maintains competitive real-time performance.

Crucially, the high-fidelity distance calculation in GEO directly translates into a superior *obstacle clearance*. For cone-shaped obstacles, GEO provides a minimum distance (*min-norm*) of $0.3239 \pm 0.1820$ m, whereas BBOX is limited to $0.2663 \pm 0.1452$ m. This additional 5.7 cm of clearance confirms that GEO effectively reduces the "dead space" overhead typical of volumetric over-approximations. By providing a tighter and more accurate robustness boundary, the framework maximizes the available shared workspace, allowing the robot

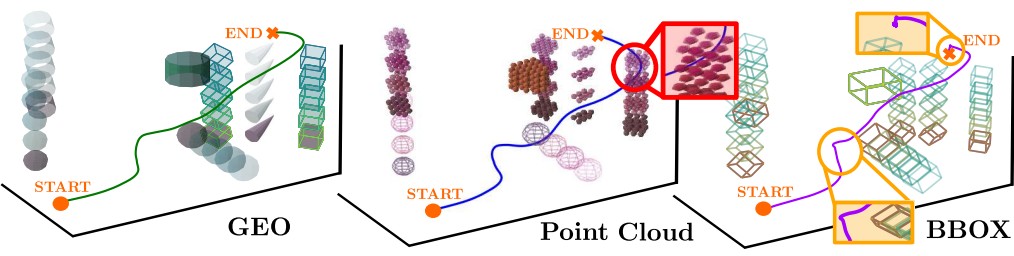
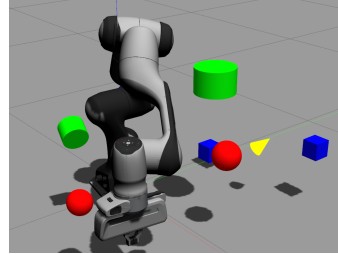

(a) Comparison of different modeling representations.

(b) Franka Emika Simulation.

Fig. 6: **Trajectory adaptation under different HRI modeling techniques.** Dynamic obstacles move toward the background (darker colors) while the central cylinder remains static. *(i) GEO (Ours)*: analytical primitives provide smooth distance fields, yielding a collision-free trajectory with faster convergence and lower jerk; *(ii) P.CLOUDS*: The discrete representation introduces local inaccuracies that lead to collision with the obstacle on the right (red inset). *(iii) BBOX*: Volumetric over-approximation and geometric unsmoothness at sharp corners lead to jerky motions (orange insets)

TABLE I: *Calculation Time* and distance to obstacles (*Obstacle Clearance*) (Mean $\pm$ SD). Each scenario was designed with an average of $12.50 \pm 3.78$ cones, $10.85 \pm 3.45$ cylinders, $12.25 \pm 5.23$ planes, or $11.90 \pm 4.08$ cubes per scene. *Total* and *Per Obj.* denote cumulative and individual calculation times, respectively. Best values are marked in blue.

| Method | Obj. | Calculation Time ($\mu s$) total / per obj. | Obstacle Clearance (m) Min-Norm |
|---|---|---|---|
| GEO | Cone | $21.9 \pm 3.1/1.8 \pm 0.4$ | $0.3239 \pm 0.1820$ |
| | Cylinder | $20.1 \pm 2.2/2.0 \pm 0.6$ | $0.2893 \pm 0.1542$ |
| | Plane | $29.2 \pm 7.5/2.7 \pm 0.9$ | $0.2806 \pm 0.1481$ |
| | Cube | $120.9 \pm 42.9/10.4 \pm 2.4$ | $0.2290 \pm 0.1419$ |
| BBOX | Cone | $118.2 \pm 45.1/9.5 \pm 2.2$ | $0.2663 \pm 0.1452$ |
| | Cylinder | $104.5 \pm 36.0/9.8 \pm 2.2$ | $0.2612 \pm 0.1478$ |
| | Plane | $112.6 \pm 47.3/11.3 \pm 1.9$ | $0.2675 \pm 0.1482$ |
| | Cube | $115.5 \pm 43.5/9.9 \pm 2.1$ | $0.2290 \pm 0.1419$ |
| P.CLOUDS | Cone | $224.8 \pm 69.7/21.3 \pm 5.8$ | $0.2682 \pm 0.1469$ |
| | Cylinder | $267.3 \pm 117.4/22.2 \pm 3.5$ | $0.2668 \pm 0.1502$ |
| | Plane | $305.0 \pm 137.3/28.3 \pm 5.2$ | $0.2756 \pm 0.1486$ |
| | Cube | $1640.8 \pm 689.3/140.3 \pm 33.3$ | $0.2206 \pm 0.1430$ |

TABLE II: Quantitative evaluation of 15 common success seeds (Mean $\pm$ STD). Best values are marked in blue.

| Metric | BBOX | GEO (Ours) | P.CLOUDS |
|---|---|---|---|
| Max Tilt Error (deg) $\downarrow$ | $0.84 \pm 0.26$ | $0.73 \pm 0.26$ | $0.89 \pm 0.31$ |
| Min Distance (mm) $\uparrow$ | $9.4 \pm 3.5$ | $11.6 \pm 8.4$ | $9.1 \pm 0.9$ |
| Time to $\leq 0.01$m (s) $\downarrow$ | $3.8 \pm 0.9$ | $3.4 \pm 0.8$ | $4.2 \pm 1.9$ |
| Path to $\leq 0.01$m (m) $\downarrow$ | $0.91 \pm 0.12$ | $0.85 \pm 0.12$ | $0.92 \pm 0.12$ |
| Max Jerk ($10^2$m/s$^3$) $\downarrow$ | $6.07 \pm 2.25$ | $4.88 \pm 1.92$ | $5.09 \pm 2.36$ |
| Max Acc ($10^2$m/s$^2$) | $90.9 \pm 24.6$ | $90.7 \pm 29.1$ | $87.8 \pm 26.5$ |

to navigate closer to obstacles without triggering conservative emergency stops.

*b) Motion Quality and Path Optimality:* To evaluate the impact of geometric modelling on task execution, we tested with 100 random seeds and analyzed a subset of 15 common success seeds (Table II). Under the attraction-based planner, GEO consistently generates smoother and natural trajectories with shorter path lengths compared to BBOX and P.CLOUDS, a distinction clearly visible in the comparison between Fig. 6.

Our method reaches the goal region ($\leq 0.01$ m) in $3.4 \pm 0.8$ s, representing a 10.5% improvement in convergence speed over BBOX ($3.8 \pm 0.9$ s) and a 19% improvement over P.CLOUDS ($4.2 \pm 1.9$ s). This agility is paired with a shorter path length ($0.85 \pm 0.12$ m), indicating that the smooth analytical gradients provided by the GEO module allow the robot to identify more direct paths through cluttered spaces.

Interestingly, GEO exhibits superior dynamic smoothness, recording the lowest *maximum jerk* in average and standard deviation $(4.88 \pm 1.92) \times 10^2$ m/s$^3$, which represents a *19.6% reduction* compared to the BBOX baseline. As shown in the zoomed-in insets of Fig. 6(a), the geometric unsmoothness of BBOX at sharp corners leads to significant oscillations. This reduction in jerk further supports the stability required for liquid transportation, by minimizing high-frequency oscillations. Moreover, smoother profiles align with our objectives to improve HRI while transporting liquids, as it is well-known that minimum-jerk motions are perceived as more natural in human motor behavior [39].

TABLE III: User-study results: median [IQR] and non-parametric analysis under a 6-pt likert scale (0-5) across three tasks (i) teleoperation with/w.o SFS; (ii) with/w.o HC; and (iii) physical interaction (PC).

| Metric | HANDLE | Baseline | $p_W$ | $p_H$ | Effect ($r$) |
|---|---|---|---|---|---|
| | **w. Slosh-free** | **w.o. Slosh-free** | | | |
| Ease of Operation | $4.0[4, 5]$ | $3.0[1, 3]$ | 0.005 | 0.024 | 0.85 |
| Perceived Slosh-free Saf. | $5.0[4, 5]$ | $2.0[0, 3]$ | 0.001 | 0.009 | 0.99 |
| | **w. Env.Robust** | **w.o. Env.Robust** | | | |
| Ease of Operation | $4.0[4, 5]$ | $4.0[4, 5]$ | 0.863 | 1.000 | 0.04 |
| Perceived Human robustness | $5.0[4, 5]$ | $1.0[0, 2]$ | $< 0.001$ | 0.009 | 0.87 |
| Perceived Object robustness | $5.0[4, 5]$ | $1.0[1, 2]$ | $< 0.001$ | 0.009 | 0.86 |
| | **w. Phy.Contact** | **w.o. Phy.Contact** | | | |
| Ease of Operation | $5.0[4, 5]$ | $3.0[0, 4]$ | 0.003 | 0.019 | 0.76 |
| Perceived System robustness | $5.0[5, 5]$ | $2.0[1, 3]$ | 0.001 | 0.010 | 0.83 |
| Perceived Self robustness | $5.0[4, 5]$ | $3.0[1, 3]$ | 0.002 | 0.017 | 0.78 |
| Perceived Physical Effort | $1.0[1, 2]$ | $4.0[2, 5]$ | 0.012 | 0.037 | 0.65 |
| *Overall Assess* | **Full System** | **Baseline** | | | |
| Perc. motion naturalness | $4.0[4, 5]$ | $2.0[1, 4]$ | 0.007 | 0.028 | 0.70 |
| Perceived Relevance | $4.5[4, 4.7]$ | $4.3[4, 4.8]$ | 0.680 | 1.000 | 0.11 |
| Intended use | 10/15 (Agree) | 5/15 (Likely Agree) | | | |
| Perceived Necessity | 14/15 (Agree) | 1/15 (Likely Agree) | | | |