# OpenReview forum: "HANDLE: Robust Non-prehensile Liquid Manipulation in Cluttered, Human-centric Environments with Physical HRI"
_IEEE.org/ICRA/2026/Workshop/Manipulation_Robustness — ICRA 2026_

### Official Review · Reviewer_LU9v · 2026-05-03
**Review: HANDLE: Robust Non-prehensile Liquid Manipulation in Cluttered, Human-centric Environments with Physical HRI**

**Rating:** 8
**Confidence:** 4

**Review:**

Strengths:
The paper provides a clear and understandable explanation of the proposed method.
It addresses a relevant problem in robust robot manipulation, with practical relevance for real-world applications such as household robotics. The inclusion of a user study strengthens the evaluation. The experimental results are interesting and provide useful insights into the proposed approach.

Weaknesses:
The structure of Section III could be improved, as the current paragraph and subsection organization is somewhat confusing. In particular, the section appears to contain only one paragraph and one subsection, which could be streamlined for better readability.
The caption of Figure 2 still contains an author comment in blue, which should be removed before publication.

Overall Comments:
Overall, this is a strong workshop paper with a clearly presented method and meaningful evaluation.
I recommend acceptance, with minor revisions to improve formatting and presentation.

---

### Decision · Program_Chairs · 2026-05-21

Accept